# Redesigning the Transformer Architecture with Insights from Multi-particle Dynamical Systems

**Subhabrata Dutta**
Jadavpur University
India
subha0009@gmail.com

**Tanya Gautam**
IIIT-Delhi
India
tanya18048@iiitd.ac.in

**Soumen Chakrabarti**
IIT-Bombay
India
soumen@cse.iitb.ac.in

**Tanmoy Chakraborty**
IIIT-Delhi
India
tanmoy@iiitd.ac.in

## Abstract

The Transformer and its variants have been proven to be efficient sequence learners in many different domains. Despite their staggering success, a critical issue has been the enormous number of parameters that must be trained (ranging from $10^7$ to $10^{11}$) along with the quadratic complexity of dot-product attention. In this work, we investigate the problem of approximating the two central components of the Transformer — multi-head self-attention and point-wise feed-forward transformation, with reduced parameter space and computational complexity. We build upon recent developments in analyzing deep neural networks as numerical solvers of ordinary differential equations. Taking advantage of an analogy between Transformer stages and the evolution of a dynamical system of multiple interacting particles, we formulate a temporal evolution scheme, `TransEvolve`, to bypass costly dot-product attention over multiple stacked layers. We perform exhaustive experiments with `TransEvolve` on well-known encoder-decoder as well as encoder-only tasks. We observe that the degree of approximation (or inversely, the degree of parameter reduction) has different effects on the performance, depending on the task. While in the encoder-decoder regime, `TransEvolve` delivers performances comparable to the original Transformer, in encoder-only tasks it consistently outperforms Transformer along with several subsequent variants. Code is available in: `https://github.com/LCS2-IIITD/TransEvolve`.

## 1 Introduction

Neural networks have evolved from early feed-forward and convolutional networks, to recurrent networks, to very deep and wide 'Transformer' networks based on attention [Vaswani et al., 2017]. Transformers and their enhancements, such as BERT [Devlin et al., 2019], T5 [Raffel et al., 2020] and GPT [Brown et al., 2020] are, by now, the default choice in many language applications. Both their training data and model sizes are massive. BERT-base has 110 million parameters. BERT-large, which often leads to better task performance, has 345 million parameters. GPT-3 has 175 billion trained parameters. Larger BERT models already approach the limits of smaller GPUs. GPT-3 is outside the resource capabilities of most research groups. Training these gargantuan models is even more challenging, with significant energy requirements and carbon emissions [Strubell et al., 2020].

In response, a growing community of researchers is focusing on post-facto reduction of model sizes, which can help with the deployment of pre-trained models in low-resource environments. However, training complexity is also critically important. A promising recent approach to faster training uses a

way of viewing layers of attention as solving ordinary differential equations (ODEs) defined over a dynamical system of interacting particles [Lu et al., 2019, Vuckovic et al., 2020]. We pursue that line of work.

Simulating particle interactions over time has a correspondence to 'executing' successive layers of the Transformer network. In the forward pass at successive layers, the self-attention and position-wise feed-forward operations of Transformer correspond to computing the new particle states from the previous ones. However, the numeric function learned by the $i$-th attention layer has zero knowledge regarding the one learned by the $(i-1)$-th layer. This is counter-intuitive due to the fact that the whole evolution is temporal in nature, and this independence leads to growing numbers of trainable parameters and computing steps. We seek to develop time-evolution functionals from the initial condition alone. Such maps can then approximate the underlying ODE from parametric functions of time (the analog of network depth) and do not require computing self-attention over and over.

We propose such a scheme, leading to a network/method we call `TransEvolve`. It can be used for both encoder-decoder and encoder-only applications. We experiment on several tasks: neural machine translation, whole-sequence classification, and long sequence analysis with different degrees of time-evolution. `TransEvolve` outperforms Transformer base model on WMT 2014 English-to-French translation by $1.4$ BLEU score while using $10\%$ fewer trainable parameters. On all the encoder-only tasks, `TransEvolve` outperforms Transformer, as well as several strong baselines, with $50\%$ fewer trainable parameters and more than $3\times$ training speedup.

## 2   Related Work

Our work focuses on two primary areas of machine learning — understanding neural networks as dynamical systems and bringing down the overhead of Transformer-based models in terms of training computation and parameters.

**Neural networks and dynamical systems.** Weinan [2017] first proposed that machine learning systems can be viewed as modeling ordinary differential equations describing dynamical systems. Chang et al. [2018] explored this perspective to analyze deep residual networks. Ruthotto and Haber [2019] later extended this idea with partial differential equations. Lu et al. [2018] showed that any parametric ODE solver can be conceptualized as a deep learning framework with infinite depth. Chen et al. [2018] achieved ResNet-comparable results with a drastically lower number of parameters and memory complexity by parameterizing hidden layer derivatives and using ODE solvers. Many previous approaches applied sophisticated numerical methods for ODE approximation to build better neural networks [Haber and Ruthotto, 2017, Zhu and Fu, 2018]. Vuckovic et al. [2020] developed a mathematical formulation of self-attention as multiple interacting particles using a measure-theoretic perspective. The very first attempt to draw analogies between Transformers and dynamical systems was made by Lu et al. [2019]. They conceptualized Transformer as a numerical approximation of dynamical systems of interacting particles. However, they focused on a better approximation of the ODE with a more robust splitting scheme (with the same model size as Transformer and the dot-product attention kept intact). We seek to parameterize the temporal dynamics to bypass attention up to a certain degree.

**Efficient variations of Transformer.** Multiple approaches have been put forward to overcome the quadratic complexity of Transformers [Wang et al., 2020a, Choromanski et al., 2020, Peng et al., 2021, Xiong et al., 2021, Liu et al., 2018]. Kitaev et al. [2020] sought to use locality-sensitive hashing and reversible residual connections to deal with long range inputs. Some studies explored the sparsified attention operations to decrease computation cost [Liu et al., 2018, Ho et al., 2019, Roy et al., 2021]. These sparsification tricks can be done based on the data relevant to the task [Roy et al., 2021, Sukhbaatar et al., 2019] or in a generic manner [Liu et al., 2018, Ho et al., 2019]. Wang et al. [2020a] observed self-attention to be low-rank, and approximated an SVD decomposition of attention to linearize it. Peng et al. [2021] used random feature kernels to approximate the softmax operation on the attention matrix. Choromanski et al. [2020] sought to linearize Transformers without any prior assumption of low-rank or sparse distribution, using positive orthogonal random features. Lee-Thorp et al. [2021] achieved remarkable speedup over Transformers by substituting the attention operation with an unparameterized Fourier transform. A bulk of these works are suitable for encoder-only tasks and often incurs slower training (e.g, Peng et al. [2021] observed $15\%$ slower training compared to Transformer). Our method overcomes both of these drawbacks.

Table 1: List of important notations and their denotations used.

| Notation | Denotation |
|---|---|
| $d, d'$ | Hidden and temporal dimension of the model |
| $\mathbf{X}^l$ | Sequence of $d$-dimensional vectors input to the $l$-th encoder block |
| $T^l$ | $d'$-dimensional map of depth (time) $l$ |
| $\mathbf{H}^l$ | Output of softmax attention at $l$-th encoder block |
| $W_q, W_k$ | Query and key projection matrices |
| $\tilde{W}_q, \tilde{W}_k$ | Temporal query and key projection matrices |
| $W_o$ | Attention output projection matrix |
| $\mathbf{A}_0$ | Query-key dot-product from initial values |
| $\mathbf{A}_1, \mathbf{A}_2, A_3$ | Time-evolution operators for attention |
| $U^l, V^l$ | Random rotation matrices at depth $l$ |

**Transformer pruning and compression.** Knowledge distillation has been used to build light-weight *student* model from large, trained Transformer models [Behnke and Heafield, 2020, Sanh et al., 2019, Wang et al., 2020b]. Michel et al. [2019] experimented with pruning different attention heads of BERT to observe redundancy in computation. However, these methods still require a trained, parameter-heavy model to start with. Tensorization approach has shown efficient compression of Transformer-based language models [Ma et al., 2019, Khrulkov et al., 2019].

## 3 Transformers as Dynamical Systems

A single block of the Transformer encoder [Vaswani et al., 2017] is defined as a multi-head self-attention layer followed by two feed-forward layers, along with residual connections. The $j$-th head of self-attention operation in the $l$-th encoder block, with $j \in \{1, \ldots, m\}$, on a given length-$n$ sequence of $d$ dimensional input vectors $\mathbf{X}^l := \{X_i^l | X_i^l \in \mathbb{R}^d\}_{i=1}^n$ can be defined as:

$$H_j^l = \text{Softmax}_i((\mathbf{X}^l W_q^l)(\mathbf{X}^l W_k^l)^\top / \sqrt{d_k})(\mathbf{X}^l W_v^l) \tag{1}$$

where $W_q^l, W_k^l \in \mathbb{R}^{d \times d_k}$, and $W_v^l \in \mathbb{R}^{d \times d_v}$ are linear projection layers and $d_k = d/m$. Conventionally, $d_v = d_k$. Each $H_j^l$ is aggregated to produce the output of the multi-head self-attention as follows:

$$\mathbf{H}^l = \text{Concat}(\{H_j^l\}_{j=1}^m)W_o^l + \mathbf{X}^l \tag{2}$$

where $W_o \in \mathbf{R}^{d \times d}$ is a linear projection. The subsequent feed-forward transformation can be defined as:

$$\mathbf{X}^{l+1} = (\sigma(\mathbf{H}^l W_{ff1}^l + B_{ff1}^l))W_{ff2}^l + B_{ff2}^l + \mathbf{H}^l \tag{3}$$

where $W_{ff1}^l \in \mathbb{R}^{d \times d_{ff}}$, $W_{ff2}^l \in \mathbb{R}^{d_{ff} \times d}$, $B_{ff1}^l \in \mathbb{R}^{d_{ff}}$, $B_{ff2}^l \in \mathbb{R}^d$, and $\sigma()$ is a non-linearity (Relu in [Vaswani et al., 2017] or Gelu in [Devlin et al., 2019]).

As Lu et al. [2019] argued, Equations 1-3 bear a striking resemblance with systems of interacting particles. Given the positions of a set of interacting particles as $\mathbf{x}(t) = \{x_i(t)\}_{i=1}^n$, the temporal evolution of such a system is denoted by the following ODE:

$$\frac{d}{dt}x_i(t) = F(x_i(t), \mathbf{x}(t), t) + G(x_i(t), t) \tag{4}$$

with the initial condition $x_i(t_0) = s_i \in \mathbb{R}^d$. The functions $F$ and $G$ are often called the *diffusion* and *convection* functions, respectively — the former models the inter-dependencies between the particles at time $t$, while the latter models the independent dynamics of each particle. Analytical solution of such an ODE is often impossible to find, and the most common approach is to use numerical approximation over discrete time intervals $[t_0, t_0 + \delta t, \ldots, t_0 + L\delta t]$. Following Euler's method of first order approximation and the Lie-Trotter splitting scheme, one can approach the numerical solution of Equation 4 from time $t$ to $t + \delta t$ as:

$$\tilde{x}_i(t) = x_i(t) + \delta t F(x_i(t), \mathbf{x}(t), t) = x_i(t) + \mathcal{F}(x_i(t), \mathbf{x}(t), t)$$
$$x_i(t + \delta t) = \tilde{x}_i(t) + \delta t G(\tilde{x}_i(t), t) = \tilde{x}_i(t) + \mathcal{G}(\tilde{x}_i(t), t) \tag{5}$$

Equation 5 can be directly mapped to the operations of the Transformer encoder given in Equations 1-3, with $\mathbf{X}^l \equiv \mathbf{x}(t)$, $\mathbf{H}^l \equiv \{\tilde{x}_i(t)\}_{i=1}^n$ and $\mathbf{X}^{l+1} \equiv \mathbf{x}(t + \delta t)$. $\mathcal{F}(\cdot, \cdot, t)$ is instantiated by the

projections $W_q^l, W_k^l, W_v^l, W_o^l$ and $\text{Softmax}(\cdot)$ operations in the multi-head self-attention. $\mathcal{G}(\cdot, t)$ corresponds to the projections $W_{ff1}^l, W_{ff2}^l, B_{ff1}^l, B_{ff1}^l$ and the $\sigma(\cdot)$ non-linearity in Equation 3.

From the described analogies, it quickly follows that successive multi-head self-attention and the point-wise feed-forward operations follow a temporal evolution (here time is equivalent to the depth of the encoder). However, Transformer and its subsequent variants parameterize these two functions in each layer separately. This leads to a large number of parameters to be trained (ranging from $7 \times 10^7$ in neural machine translation tasks to $\sim 175 \times 10^9$ in language models like GPT-3). We proceed to investigate how one can leverage the temporal evolution of the diffusion and convection components to bypass this computational bottleneck.

## 4   Time-evolving Attention

As Equation 5 computes $\mathbf{x}(t)$ iteratively from a given initial condition $\mathbf{x}(t_0) = \mathbf{s} = \{s_i\}_{i=1}^n$, one can reformulate the diffusion map $\mathcal{F}(\cdot, \mathbf{x}(t), t)$ as $\tilde{\mathcal{F}}(\cdot, f(\mathbf{s}, t))$, i.e., as a functional of the initial condition and time only. When translated to the case of Transformers, this means one can avoid computing pairwise dot-product between $n$ input vectors at each layer by computing a functional form at the beginning and evolving it in a temporal (depth-wise) manner. We derive this by applying dot-product self-attention on hypothetical input vectors with augmented depth information, as follows.

Let $\mathbf{X}' = \{X_i' | X_i' \in \mathbb{R}^{d+d'}\}$ be a set of vectors such that $X_i' = \text{Concat}(X_i^0, T^l)$, where $X_i^0 = \{x_1^i, \dots, x_d^i\}$ and $T^l = \{\tau_1(l), \dots, \tau_{d'}(l)\}$. $W_q', W_k' \in \mathbb{R}^{(d+d') \times (d+d')}$ are the augmented query and key projections, respectively, such that $W_q' = [\omega_{ij}]_{i,j=1,1}^{d+d',d+d'}$, $W_k' = [\theta_{ij}]_{i,j=1,1}^{d+d',d+d'}$. The pre-softmax query-key dot product between $X_i'$ and $X_j'$ is given by $a_{ij}' = (X_i' W_q')(X_j' W_k')^\top$. We can decompose $W_q'$ as concatenation of two matrices $W_q, \tilde{W}_q$ such that $W_q = [\omega_{ij}]_{i,j=1,1}^{d,d+d'}$ and $\tilde{W}_q = [\omega_{ij}]_{i,j=d+1,1}^{d+d',d+d'}$. Similarly, we decompose $W_k'$ into $W_k$ and $\tilde{W}_k$. Then $a_{ij}'$ can be re-written as:

$$
\begin{aligned}
a_{ij}' &= (X_i^0 W_q)(X_j^0 W_k)^\top + (X_i^0 W_q)(T^l \tilde{W}_k)^\top + (T^l \tilde{W}_q)(X_j^0 W_k)^\top + (T^l \tilde{W}_q)(T^l \tilde{W}_k)^\top \\
&= a_{ij}^0 + A_{1i} T^{l\top} + T^l A_{2j} + A_3 (T^l \odot T^l)
\end{aligned}
\tag{6}
$$

where $A_{1i} = X_i^0 W_q \tilde{W}_k^\top$, $A_{2j} = \tilde{W}_q W_k^\top X_j^{0\top}$, $A_3 = \tilde{W}_q \tilde{W}_k^\top$ and $\odot$ signify hadamard product. Detailed derivation is provided in Appendix A.

It is evident that $a_{ij}^0$ is the usual dot-product pre-softmax attention between vector elements $X_i^0, X_j^0$. For the complete sequence of vector elements $\mathbf{X}$, we write $\mathbf{A}_0 = [a_{ij}]_{ij}$, $\mathbf{A}_1 = \{A_{1i}\}_i$ and $\mathbf{A}_2 = \{A_{2j}\}_j$. By definition, $T^l$ is a vector function of the depth $l$. To construct $T^l$ as a vector function of $l$, we formulate

$$
T^l = \left[ w_1^l \sin(\frac{l}{P}), \dots, w_{\frac{d'}{2}}^l \sin(\frac{d'l}{2P}), w_{\frac{d'}{2}+1}^l \cos(\frac{l}{P}), \dots, w_{d'}^l \cos(\frac{d'l}{2P}) \right]
\tag{7}
$$

where $W_t^l = \{w_i^l\}_{i=1}^{d'}$ are learnable parameters at depth $l$, and $P = \frac{d'L}{2\pi}$. Such a choice of $T^l$ is intuitive in the following sense: let $C = AT^l + B$ for some arbitrary $A = [a_{ij}] \in \mathbb{R}^{p \times d}$, $B = \{b_i\} \in \mathbb{R}^p$, and $C = \{c_i\} \in \mathbb{R}^p$. Then we get

$$
c_i = b_i + \sum_{j=1}^{\frac{d'}{2}} a_{ij} w_j^l \sin(\frac{jl}{P}) + \sum_{j=1}^{\frac{d'}{2}} a_{ij} w_{j+\frac{d'}{2}}^l \cos(\frac{jl}{P})
\tag{8}
$$

In other words, any feed-forward transformation on $T^l$ gives us a vector in which each component is a Fourier series (thereby, approximating arbitrary periodic functions of $l$ with period $P$). This enables us to encode the nonlinear transformations that $\mathbf{X}$ undergoes in successive blocks. With this, $\mathbf{A}_1, \mathbf{A}_2, A_3$ constitute the time-evolution operators to map depth information to the attention, computed only from the initial conditions $X_i^0, X_j^0$.

Figures 1(a) and 1(b) summarize the procedure. Given two input sequences $\mathbf{X}$ and $\mathbf{Y}$ (in case of self-attention, they are the same), we first compute $\mathbf{A}_0, \mathbf{A}_1, \mathbf{A}_2, A_3$ using the linear projection matrices $W_q, W_k, \tilde{W}_q, \tilde{W}_k$, as described above. Additionally, we normalize $\mathbf{A}_0$ by a factor $\frac{1}{\sqrt{d/m}}$, where $m$ is the number of heads, similar to Transformer. Then, at any subsequent layer $l \in \{1, \dots, L\}$, instead

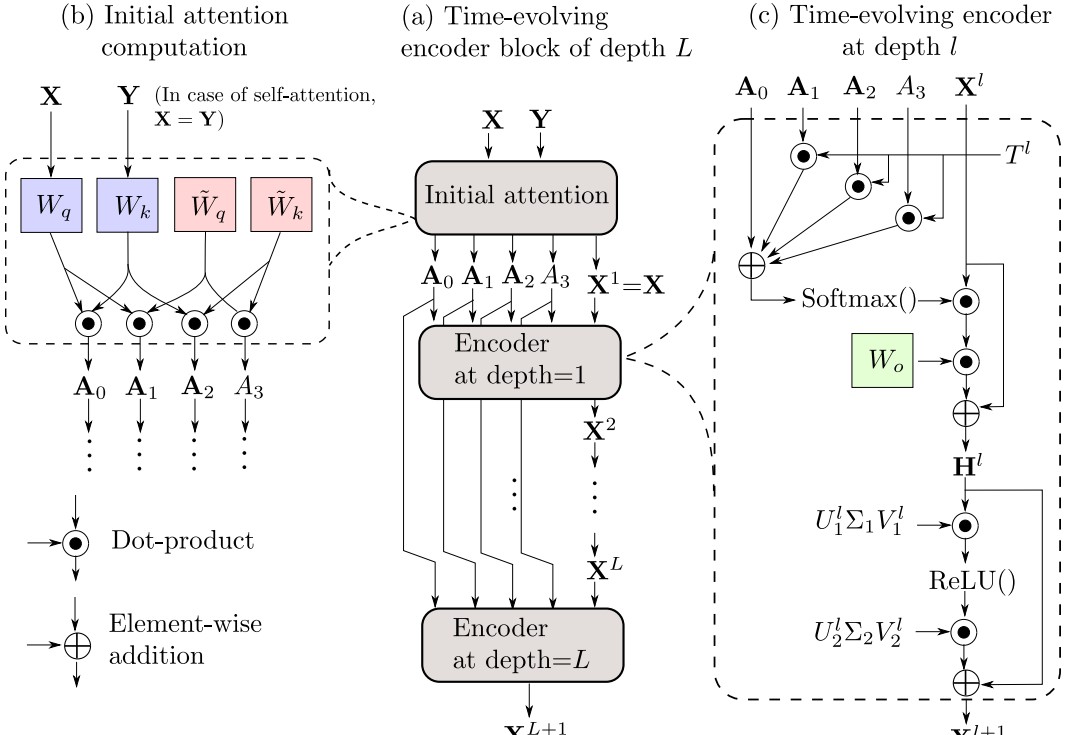

(b) Initial attention computation

(a) Time-evolving encoder block of depth $L$

(c) Time-evolving encoder at depth $l$

Dot-product

Element-wise addition

Figure 1: Dissecting the primary functional components of `TransEvolve`. (a) An $L$-depth encoder block starts with computing (b) the initial condition matrix $\mathbf{A}_0$ and the evolution operator matrices $\mathbf{A}_1, \mathbf{A}_2, A_3$ from the input sequence. These four are then used in (c) each encoder at depth $l$ along with a vector function of depth, $T^l$ to apply the attention operation on the output from the previous step. This product of attention is then passed to the feed-forward transformation actuated by the depth-dependent, random rotation matrices $U_1^l, U_2^l, V_1^l, V_2^l$ (`TransEvolve`-**randomFF**, see Section 6). In another variation, we use learnable feed-forward layers (`TransEvolve`-**fullFF**).

of computing the query, key and value projections from the input $\mathbf{X}^l$, we use the previously computed $\mathbf{A}_0, \mathbf{A}_1, \mathbf{A}_2, A_3$ along with $T^l$. Then the time-evolved attention operation becomes

$$\mathbf{H}^l = \text{Softmax}(\mathbf{A}_0 + \mathbf{A}_1 T^{l\top} + T^l \mathbf{A}_2 + A_3 (T^l \odot T^l)) \mathbf{X}^l W_o^l + \mathbf{X}^l \qquad (9)$$

For the sake of brevity, we have not shown the concatenation of multi-headed splits as shown in Equation 2. Therefore, one should identify $W_o^l$ in Equation 9 with Equation 2 and not as value projection as in Equation 1. We do not use value projection in our method. Also, in the multi-headed case, all the matrix-vector dot products shown in Equation 9 are computed on head-splitted dimension of size $d/m$.

**Complexity and parameter reduction.** Given a $d$-dimensional input sequence of length $n$, computing the pre-softmax attention weight matrix in a single Transformer encoder requires $\mathcal{O}(n^2 d)$ multiplications. In our proposed method, the complexity of calculating dot-product attention (corresponding to $a_{ij}^0$ in Equation 6) invokes computations of similar order – $\mathcal{O}(n^2(d + d'))$. However, this is needed only once at the beginning. The subsequent attention matrices are calculated using the components with $T$ in Equation 6. Both $\mathbf{A}_1 T$ and $T \mathbf{A}_2$ require $\mathcal{O}(nd')$ computation, and $A_3 T$ requires only $\mathcal{O}(d')$. Therefore, if one tends to use $L$ successive attention operations, our proposed method is computationally equivalent to a single original dot-product self-attention followed by multiple cheaper stages. In addition to this, attention weight computation using Equation 6 eliminates the need for query, key, and value projections at each self-attention block. Thereby, it ensures a parameter reduction of $\mathcal{O}(Ld^2)$ for a total stacking depth of $L$.

# 5 Time-evolving Feed-forward

The Transformer counterpart of the convection function $\mathcal{G}(\cdot, t)$ is the point-wise feed-forward transformation in Equation 3. The complete operation constitutes two projection operations: first, the input vectors are mapped to a higher dimensional space ($\mathbb{R}^d \to \mathbb{R}^{d_{ff}}$) along with a non-linear transformation, followed by projecting them back to their original dimensionality ($\mathbb{R}^{d_{ff}} \to \mathbb{R}^d$). At the $l$-th Transformer encoder, these dimensionality transformations are achieved by the matrices $W^l_{ff1}$ and $W^l_{ff2}$, respectively (see Equation 3). To construct their temporal evolution, we attempt to decompose them into time-evolving components.

Recall that any real $m \times n$ matrix $M$ can be decomposed as $M = U\Sigma V^\top$ where $U \in \mathbb{R}^{m \times m}$, $V \in \mathbb{R}^{n \times n}$ are orthogonal, and $\Sigma \in \mathbb{R}^{m \times n}$ is a rectangular diagonal matrix. However, computing exact orthogonal matrices with large dimensionality is computationally infeasible. Instead, we construct approximate rotation matrices $U \in \mathbb{R}^{d \times d}$ as:

$$U^l = \frac{1}{\sqrt{d}} \begin{bmatrix} \sin(w^l_{11}\frac{l}{P}) & \dots & \sin(w^l_{1\frac{d}{2}}\frac{dl}{2P}) & \cos(w^l_{11}\frac{l}{P}) & \dots & \cos(w^l_{1\frac{d}{2}}\frac{dl}{2P}) \\ \vdots & & & & & \vdots \\ \sin(w^l_{d1}\frac{l}{P}) & \dots & \sin(w^l_{d\frac{d}{2}}\frac{dl}{2P}) & \cos(w^l_{d1}\frac{l}{P}) & \dots & \cos(w^l_{d\frac{d}{2}}\frac{dl}{2P}) \end{bmatrix} \tag{10}$$

where $P = \frac{dL}{2\pi}$ and $w^l_{ij} \in \mathcal{N}(0, d^2)$. We discuss the properties of such a matrix $U^l$ in the Appendix B.

We construct four matrices $U^l_1 \in \mathbb{R}^{d \times d}$, $V^l_1 \in \mathbb{R}^{d_{ff} \times d_{ff}}$, $U^l_2 \in \mathbb{R}^{d_{ff} \times d_{ff}}$, and $V^l_2 \in \mathbb{R}^{d \times d}$ as described in Equation 10. Also, we construct two rectangular diagonal matrices $\Sigma_1 \in \mathbb{R}^{d \times d_{ff}}$ and $\Sigma_2 \in \mathbb{R}^{d_{ff} \times d}$ with learnable diagonal entries. With these matrices defined, one can reformulate the point-wise feed-forward operation (Equation 3) as:

$$\mathbf{X}^{l+1} = U^l_2 \Sigma_2 V^l_2 \sigma(U^l_1 \Sigma_1 V^l_1 \mathbf{H}^l + B_1) + B_2 + \mathbf{H}^l \tag{11}$$

This reformulation reduces the number of trainable parameters from $\mathcal{O}(dd_{ff})$ to $\mathcal{O}(d)$.

# 6 Proposed Model: `TransEvolve`

Equipped with the particle evolution based definitions of attention and point-wise feed-forward operations, we proceed to define the combined architecture of `TransEvolve`. The primary building blocks of our model are the initial attention computations following Equation 6, shown in Figure 1(b) and attention operation at depth $l$ followed by feed-forward transformations (shown together as the encoder at depth $l$ in Figure 1(c)). An $L$-depth encoder block of `TransEvolve`, as shown in Figure 1(a), consists of $L$ number of attention and feed-forward operations stacked successively, preceded by an initial attention computation.

**Variations in temporal evolution of feed-forward operation.** While the re-parameterization of the point-wise feed-forward operations described in Section 5 seems to provide an astonishing reduction in the parameter size, it is natural to allow different degrees of trainable parameters for these operations. This allows us to explore the effects of time-evolving approximations of attention and point-wise feed-forward separately. We design two variations of `TransEvolve`. In the original setting, the point-wise feed-forward operation is applied using random rotation matrices, following Equation 11. We call this `TransEvolve`-**randomFF**. In the other variation, the time evolution process is applied for the attention operations only while the feed-forward operations are kept to be the same as Transformer (Equation 3). This variation is denoted as `TransEvolve`-**fullFF**, henceforth.

**Different degrees of temporal evolution.** Recall from Section 3 that Equation 5 is a numerical approximation of Equation 4 so as to evaluate $\mathbf{x}(t+\delta t)$ iteratively from $\mathbf{x}(t)$. When we attempt to approximate $F(x_i(t), \mathbf{x}(t), t)$ as $\tilde{F}(x_i(t), f(\mathbf{x}(t_0), t))$, the error $|F(x_i(t), \mathbf{x}(t), t) - \tilde{F}(x_i(t), f(\mathbf{x}(t_0), t))|$ is expected to grow as $|t - t_0|$ grows. However, it is not feasible to compute the exact approximation error due to the complex dynamics which, in turn, varies from task to task. To compare with the original Transformer, we seek to explore this empirically. More precisely, the $L$-depth encoder (decoder) block with our proposed evolution strategy is expected to approximate $L$ encoder (decoder) layers of Transformer. Following the approximation error argument, `TransEvolve` is likely to bifurcate more from Transformer as $L$ gets larger. We experiment with multiple values of $L$ while keeping the total number of attention and feed-forward operations fixed (same as Transformer in the comparative task). To illustrate, for WMT English-to-German translation task, Transformer uses 6 encoder blocks

followed by 6 decoder blocks. Converted in our setting, one can use 1 encoder (decoder) each with depth $L = 6$ or 2 encoders (decoders) each with depth $L = 3$ (given that the latter requires more parameters compared to the former).

**Encoder-decoder.** To help compare with the original Transformer, we keep our design choices similar to Vaswani et al. [2017]: embedding layer tied to the output logit layer, sinusoidal position embeddings, layer normalization, etc. The temporal evolution of self-attention described in Section 4 can be straightforwardly extended to encoder-decoder attention. Given the decoder input $\mathbf{X}_{dec}^0$ and the encoder output $\mathbf{X}_{enc}$, we compute the initial dot-product attention matrices $\mathbf{A}_0^{dec}$ and $\mathbf{A}_0^{ed}$, corresponding to decoder self-attention and encoder-decoder attention, as follows:

$$\mathbf{A}_0^{dec} = (\mathbf{X}_{dec}^0 W_q^{dec})^\top (\mathbf{X}_{dec}^0 W_k^{dec}); \ \mathbf{A}_0^{ed} = (\mathbf{X}_{dec}^0 W_q^{ed})^\top (\mathbf{X}_{enc} W_k^{ed}) \tag{12}$$

If $\mathbf{X}_{dec}^0$ and $\mathbf{X}_{enc}$ are sequences of length $n_{dec}$ and $n_{enc}$, respectively, then $\mathbf{A}_0^{dec}$ and $\mathbf{A}_0^{ed}$ are $n_{dec} \times n_{dec}$ and $n_{enc} \times n_{dec}$ matrices, respectively. We also compute the corresponding time-evolution operators $\mathbf{A}_1^{dec}, \mathbf{A}_2^{dec}, A_3^{dec}$ and $\mathbf{A}_1^{ed}, \mathbf{A}_2^{ed}, A_3^{ed}$ similar to Equation 6. Then at each depth $l$ in the decoder block, the time-evolved attention operations are as follows:

$$\begin{aligned} \tilde{\mathbf{H}}^l &= \text{Softmax}(\mathbf{A}_0^{dec} + \mathbf{A}_1^{dec} T_{dec}^l + T_{dec}^l \mathbf{A}_2^{dec} + A_3^{dec}(T_{dec}^l \odot T_{dec}^l)) \mathbf{X}_{dec}^l W_o^{l,dec} + \mathbf{X}_{dec}^l \\ \mathbf{H}^l &= \text{Softmax}(\mathbf{A}_0^{ed} + \mathbf{A}_1^{ed} T_{ed}^l + T_{ed}^l \mathbf{A}_2^{ed} + A_3^{ed}(T_{ed}^l \odot T_{ed}^l)) \mathbf{X}_{enc} W_o^{l,ed} + \tilde{\mathbf{H}}^l \end{aligned} \tag{13}$$

where $T_{dec}^l$ and $T_{ed}^l$ are two independent vector maps of depth $l$ representing the time-evolutions of the decoder self-attention and the encoder-decoder attention. For the sake of brevity, we have shown the full multi-head attentions in Equations 12 and 13 without showing the concatenation operation (as in Equation 2).

**Encoder-only.** For many-to-one mapping tasks (e.g., text classification), we only use the encoder part of `TransEvolve`. The sequential representations returned from the encoder are applied with an average pooling along the sequence dimension and passed through a normalization layer to the final feed-forward layer to predict the classes.

## 7 Experiments

### 7.1 Model Configurations

We set $d = d'$ across all the variations of `TransEvolve`. For the encoder-decoder models, we experiment with the base version of `TransEvolve` with $d = 512$. For encoder-only tasks, we use a small version with $d = 256$. In both base and small versions, the number of heads is set to 8.

As discussed in Section 6, one can vary the degree of temporal evolution (and the number of trainable parameters) in `TransEvolve` by changing the value of $L$ in the $L$-depth time-evolving encoder (decoder) blocks. To keep the total depth of the model constant, we need to change the number of these blocks as well. We design two such variations. Recall that the total depth of the Transformer encoder-decoder model is 12 (6 encoders and 6 decoders); in `TransEvolve`, we choose (i) 1 encoder (decoder) block with depth 6 (denoted as `TransEvolve-fullFF-1`, `TransEvolve-randomFF-1`, etc.), and (ii) 2 encoder (decoder) blocks each with depth 3 (denoted by `TransEvolve-randomFF-2`, etc.). This way, we obtain 4 variations of `TransEvolve` for our experiments.

### 7.2 Tasks

We evaluate `TransEvolve` over three different areas of sequence learning from texts: (i) sequence-to-sequence mapping in terms of machine translation, (ii) sequence classification, and (iii) long sequence learning. The former task requires the encoder-decoder architecture while the latter two are encoder-only.

**Machine Translation (MT).** As a sequence-to-sequence learning benchmark, we use WMT 2014 English-to-German (En-De) and English-to-French (En-Fr) translation tasks. The training data for these two tasks contain about $4.5$ and $35$ million sentence pairs, respectively. For both these tasks, we use the base configuration of our proposed models. We report the performance of the En-De model on WMT 2013 and 2014 En-De test sets (*newstest2013* and *newstest2014*). En-Fr model is tested only on WMT 2014 test set. Implementation details on these tasks are described in the Appendix.

**Sequence classification.** We evaluate the performance of the encoder-only version of our model on two text classification datasets: **IMDB movie-review** dataset [Maas et al., 2011] and **AGnews** topic

| Model | En-De | | En-Fr | #Params |
|---|---|---|---|---|
| | WMT 2013 | WMT 2014 | WMT 2014 | |
| Transformer BASE | 25.8 | **27.3** | 38.1 | 65M |
| `TransEvolve`- randomFF-1 | 22.5 | 23.1 | 32.6 | 27M |
| `TransEvolve`- randomFF-2 | 24.2 | 23.8 | 33.4 | 33M |
| `TransEvolve`- fullFF-1 | 25.3 | 25.8 | 38.0 | 53M |
| `TransEvolve`- fullFF-2 | **26.2** | *27.2* | **39.5** | 59M |

Table 2: Performance of `TransEvolve` variants on English-to-German (En-De) and English-to-French (En-Fr) translations in terms of BLEU scores. Transformer results are taken from the original paper [Vaswani et al., 2017].

classification dataset Zhang et al. [2015]. The IMDB dataset consists of 25000 movie reviews each for training and testing purposes. This is a binary classification task (positive/negative reviews). The AGnews dataset is a collection of news articles categorized into 4 classes (Science & Technology, Sports, Business, and World), each with 30000 and 1900 instances for training and testing, respectively. For both these tasks, we use the small version of our model. Detailed configurations and hyperparameters are given in the Appendix.

**Long sequence learning.** To test the effectiveness of our model for handling long sequences, we use two sequence classification tasks: character-level classification of the IMDB reviews and latent tree learning from sequences of arithmetic operations and operands with the **ListOps** dataset [Nangia and Bowman, 2018]. For the IMDB reviews, we set the maximum number of tokens (characters) in the training and testing examples to 4000 following [Peng et al., 2021]. In the ListOps dataset, an input sequence of arithmetic operation symbols and digits in the range 0-9 is given as input; it is a 10-way classification task of predicting the single-digit output of the input operation sequence. We consider sequences of length 500-2000 following [Peng et al., 2021]. Again, we use the small version of `TransEvolve` for these two tasks. Further experimental details are provided in the Appendix.

### 7.3 Training and Testing Procedure

All experiments are done on v3-8 Cloud TPU chips. We use Adam optimizer with learning rate scheduled per gradient update step as $lr = \frac{lr_{max}}{\sqrt{d}} \times \max(step^{-0.5}, warmup\_step^{-1.5} \times step)$. For the MT task, we set $lr_{max} = 1.5$ and $warmup\_step = 16000$. For the remaining two tasks, these values are set to 0.5 and 8000, respectively. For the translation tasks, we use a label smoothing coefficient $\epsilon = 0.1$. Training hyperparameters and other task-specific details are described in the Appendix. For the translation tasks, we report the BLEU scores averaged from 10 last checkpoints, each saved per 2000 update steps. For encoder-only tasks, we report the average best accuracy on five separate runs with different random seeds. Comprehensive additional results are provided in the Appendix.

## 8 Results and Discussion

**Machine Translation.** Table 2 summarizes the performance of `TransEvolve` variants against Transformer (base version) on English-German and English-French translation tasks. `TransEvolve`-**randomFF** versions perform poorly compared to **fullFF** versions.

Table 3: Text classification accuracy of `TransEvolve` variants on AGnews and IMDB dataset. Scores of Transformer, Linformer, and Synthesizer are taken from [Tay et al., 2020a].

| Model | AGnews | IMDB |
|---|---|---|
| Transformer | 88.8 | 81.3 |
| Linformer [Wang et al., 2020a] | 86.5 | 82.8 |
| Synthesizer [Tay et al., 2020a] | 89.1 | 84.6 |
| `TransEvolve`-randomFF-1 | 90.6 | 87.3 |
| `TransEvolve`-randomFF-2 | 90.8 | 87.5 |
| `TransEvolve`-fullFF-1 | **91.1** | 86.8 |
| `TransEvolve`-fullFF-2 | 90.5 | **87.6** |

However, with less than 50% of the parameters used by Transformers, they achieve above 85% performance of that of Transformer. With all random rotation matrices replaced by standard feed-forward layers, `TransEvolve` with a single 6 layers deep encoder (decoder) block performs comparably to Transformer on the En-Fr dataset. Finally, with 2 blocks of depth 3 encoders, `TransEvolve`-**fullFF**-2 outperforms Transformer on the WMT 2014 En-Fr dataset by 1.2 points BLEU score, despite having 10% lesser parameters. On the En-De translation task, this model performs comparable to Transformer, with 0.5 gain and 0.1 drops in BLEU scores on WMT 2013 and 2014 test datasets, respectively.

Table 4: Accuracy (%) of `TransEvolve` on the long range sequence classification tasks. Speed is measured w.r.t. Transformers on the character-level IMDB dataset for input sequences of length $1k$, $2k$, $3k$ and $4k$. All results except `TransEvolve` variants are taken from [Peng et al., 2021].

| Models | Tasks | | Speed | | | |
|---|---|---|---|---|---|---|
| | ListOps | charIMDB | $1k$ | $2k$ | $3k$ | $4k$ |
| Transformer | 36.4 | 64.3 | 1.0 | 1.0 | 1.0 | 1.0 |
| Linformer [Wang et al., 2020a] | 35.7 | 53.9 | **1.2** | 1.9 | 3.7 | 5.5 |
| Reformer [Kitaev et al., 2020] | 37.3 | 56.1 | 0.5 | 0.4 | 0.7 | 0.8 |
| Sinkhorn [Tay et al., 2020b] | 17.1 | 63.6 | 1.1 | 1.6 | 2.9 | 3.8 |
| Synthesizer [Tay et al., 2020a] | 37.0 | 61.7 | 1.1 | 1.2 | 2.9 | 1.4 |
| Big Bird [Zaheer et al., 2020] | 36.0 | 64.0 | 0.9 | 0.8 | 1.2 | 1.1 |
| Linear attention [Katharopoulos et al., 2020] | 16.1 | 65.9 | 1.1 | **1.9** | 3.7 | 5.6 |
| Performers [Choromanski et al., 2020] | 18.0 | 65.4 | **1.2** | **1.9** | **3.8** | **5.7** |
| Random Feature Attention (RFA) [Peng et al., 2021] | 36.8 | 66.0 | 1.1 | 1.7 | 3.4 | 5.3 |
| `TransEvolve-randomFF-1` | **43.2** | 65.3 | **1.2** | 1.3 | 1.2 | 1.2 |
| `TransEvolve-randomFF-2` | 39.1 | **66.1** | 1.1 | 1.2 | 1.2 | 1.1 |
| `TransEvolve-fullFF-1` | 42.2 | 65.7 | **1.2** | 1.2 | 1.2 | 1.1 |
| `TransEvolve-fullFF-2` | 37.8 | 65.6 | 1.1 | 1.1 | 1.0 | 1.1 |

**Text classification.** Table 3 summarizes the accuracy of `TransEvolve` on text classification tasks. It should be noted that these results are not comparable to the state-of-the-art results on these two datasets; all four models mentioned here use no pretrained word embeddings to initialize (which is essential to achieve benchmark results for these tasks, mostly due to the smaller sizes of these datasets) or extra data for training. These results provide a comparison of purely model-specific learning capabilities. Upon that, `TransEvolve`-**fullFF**-1 achieves the best performance on both datasets. For topic classification on AGnews, it scores $91.1\%$ accuracy, outperforming Synthesizer (the best baseline) by $2\%$. The improvements are even more substantial on IMDB. `TransEvolve`-**fullFF**-2 achieves an accuracy of $87.6\%$ with improvements of $3\%$ and $6.3\%$ upon Synthesizer and Transformer, respectively.

**Long sequence tasks.** `TransEvolve` shows remarkable performance in the arena of long sequence learning as well. As shown in Table 4, `TransEvolve` outperforms Transformer along with previous methods on both ListOps and character-level sentiment classification on IMDB reviews. On ListOps, `TransEvolve`-**randomFF**-1 establishes a new benchmark of $43.2\%$ accuracy — beating Reformer (existing state-of-the-art) by $4.9\%$ and Transformer by $5.8\%$. Moreover, all four versions of `Trans-Evolve` show a gain in accuracy compared to the previous models. It is to be noted that we use the small version of `TransEvolve` (with 256 hidden size) for these tasks, while Transformer uses the base version (512). So all these improvements are achieved while using $25\%$ of Transformer's parameter size. On the char-IMDB dataset, while only `TransEvolve`-**randomFF**-2 outperforms the existing state-of-the-art, i.e., Random Feature Attention (RFA), other variants of `TransEvolve` also turn out to be highly competitive. `TransEvolve`-**randomFF**-2 achieves $66.1\%$ accuracy, improving upon RFA by $0.1\%$ and Transformer by $1.8\%$. While all the variants of `TransEvolve` run faster compared to Transformer, they do not show any additional speed-up for longer sequences like RFA or Performers. This behavior is reasonable given the fact that `TransEvolve` still performs softmax on $\mathcal{O}(n^2)$ sized attention matrix at each depth. Moreover, the speedups reported in Table 4 are with the same batch size for Transformer. Practically, the lightweight `TransEvolve`-**randomFF** models can handle much larger batch size (along with larger learning rates). This results in a more than $3\times$ training speedup for all the lengths compared to Transformer. The relative gain in speed compared to Transformer or Reformer is achieved due to linear computation of pre-softmax weights (except for the initial attention calculation) and a reduced number of parameters. This is also supported by the fact that **randomFF** versions run faster than **fullFF** ones, and speed decreases with the increased number of shallower encoder blocks.

**Effects of model variations.** Random rotation matrices, instead of standard feed-forward layers and varying the number (conversely, the depth) of `TransEvolve` encoder blocks, show a task-dependent effect on performance. `TransEvolve` versions with a single 6-layer deep encoder perform better than 2 successive 3-layer deep encoders in the case of ListOps. In other encoder-only tasks, there are no clear winners among the two. Similar patterns can be observed among `TransEvolve`-**randomFF** and `TransEvolve`-**fullFF**. In the case of machine translation though, it is straightforward that having a lesser number of feed-forward parameters (**randomFF** vs. **fullFF**) and/or fewer encoder blocks

with deeper evolution deteriorate the performance. In general, the performance of `TransEvolve` variations degrades with the decrease in the total parameter size on translation tasks. Encoder-only models use only self-attention, while the encoder-decoder models use self, cross, and causal attentions. From the viewpoint of multi-particle dynamical systems, the latter two are somewhat different from the former one. In self-attention, each of the 'particles' is interacting with each other simultaneously at a specific time-step. However, in causal attention, the interactions are ordered based on the token positions even within a specific timestep. So while decoding, there is an additional evolution of token representations at each step. Moreover, in encoder-decoder tasks as well, TransEvolve outperforms the original Transformer in two datasets (En-De 2013 and En-Fr 2014) clearly, while providing comparable performance in another (En-De 2014). It is the random matrix versions that suffered the most in these versions. The way we designed the random matrix feedforward transformations incorporates the depth information via evolution while reducing the parameter size. In encoder-only tasks, this evolution scheme looks comparable to depth-independent, parameter-dense full versions in expressive power. Moreover, each of the tasks presents different learning requirements. Intuitively, ListOps or character-level sentiment classification tasks have a smaller input vocabulary (hence, fewer amounts of information to encode in each embedding), but longer intra-token dependency (resulting in a need for a powerful attention mechanism) compared to sentiment or topic classification at the word level. Random matrix versions provide depth-wise information sharing that may facilitate the model to better encode complex long-range dependencies, but they might remain under-parameterized to transform information-rich hidden representations. This complexity trade-off can be the possible reason behind randomFF versions, outperforming fullFF in both the long-range tasks while vice versa in text classification tasks.

These variations indicate that `TransEvolve` is *not just a compressed approximation of Transformer*. Particularly in the case of encoder-only tasks, the depth-wise evolution of self-attention and feed-forward projection helps `TransEvolve` to learn more useful representations of the input with a much fewer number of trainable parameters. This also suggests that *the nature of the dynamical system underlying a sequence learning problem differs heavily with different tasks*. For example, the staggering performance of `TransEvolve`-**randomFF**-1 on the ListOps dataset implies that the diffusion component of the underlying ODE is more dominant over the convection component, and the error $|F(x_i(t), \mathbf{x}(t), t) - \tilde{F}(x_i(t), f(\mathbf{x}(t_0), t))|$ remain small with increasing $|t - t_0|$. One may even conjecture whether the Hölder coefficients of the functions $F$ and $G$ (in Equation 4) underlying the learning problem govern the performance difference. However, we leave this for future exploration.

## 9    Conclusion

Transformer stacks provide a powerful neural paradigm that gives state-of-the-art prediction quality, but they come with heavy computational requirements and large models. Drawing on an analogy between representation evolution through successive Transformer layers and dynamic particle interactions through time, we use numerical ODE solution techniques to design a computational shortcut to Transformer layers. This not only lets us save trainable parameters and training time complexity, but can also improve output quality in a variety of sequence processing tasks. Apart from building a better-performing model, this novel perspective carries the potential to uncover an in-depth understanding of sequence learning in general.

**Limitations.** It is to be noted that `TransEvolve` does not get rid of quadratic operations completely, as in the case of linear attention models like Linformer or Performer. It still performs costly softmax operations on quadratic matrices. Also, at least for once (from the initial conditions) `TransEvolve` computes pre-softmax $\mathcal{O}(n^2)$ dot-product matrices. However, the significant reduction in model size compensates for this pre-existing overhead. Also, the speedup achieved by `TransEvolve` is majorly relevant in the training part and many-to-one mapping. In the case of autoregressive decoding, `Trans-Evolve` does not provide much gain over the Transformer compared to the linear versions [Peng et al., 2021]. Since the linearization approaches of the attention operation usually seeks to approximate the pair-wise attention kernel $k(x, y)$ as some (possibly random) feature map $\phi(x)\phi(y)$, one may seek to design a temporal evolution of the kernel by allowing temporally evolving feature maps. We leave this as potential future work.

## Acknowledgement

The authors would like to acknowledge the support of Tensorflow Research Cloud. for generously providing the TPU access and the Google Cloud Platform for awarding GCP credits. T. Chakraborty would like to acknowledge the support of Ramanujan Fellowship, CAI, IIIT-Delhi and ihub-Anubhuti-iiitd Foundation set up under the NM-ICPS scheme of the Department of Science and Technology, India. S. Chakrabarti is partly supported by a Jagadish Bose Fellowship and a grant from IBM.

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
