# A Derivation of time-evolving attention operators

We show the full derivation of Equation 6 as follows. Let $\mathbf{X}' = \{X_i'|X_i' \in \mathbb{R}^{d+d'}\}_{i=1}^n$ be a sequence of vectors (which is the original $d$-dimensional input augmented with $d'$-dimensional depth information). Let us further assume $X_i' = \{x_{ij}'|x_{ij}' \in \mathbb{R}\}_{j=1}^{d+d'}$. For two projection matrices $W_q', W_k' \in \mathbb{R}^{d \times (d+d')}$ where $W_q' = [\omega_{ij}]_{i,j=1}^{d+d',d+d'}$ and $W_k' = [\theta_{ij}]_{i,j=1}^{d+d',d+d'}$, the query and key projections become:

$$Q_i = X_i'W_q' = \{q_{ij}|q_{ij} = \sum_{l=1}^{d+d'} x_{il}'\omega_{lj}\}_{j=1}^{d+d'}$$

$$K_i = X_i'W_k' = \{k_{ij}|q_{ij} = \sum_{l=1}^{d+d'} x_{il}'\theta_{lj}\}_{j=1}^{d+d'}$$

Then, the pre-softmax dot-product attention matrix for $\mathbf{X}'$ becomes $\mathbf{A}' = [a_{ij}']_{i,j=1}^{n,n}$ where

$$a_{ij}' = Q_iK_j = \sum_{\alpha=1}^{d+d'} (q_{i\alpha}k_{j\alpha})$$

$$= \sum_{\alpha=1}^{d+d'} \left( \sum_{\beta=1}^{d+d'} x_{i\beta}'\omega_{\beta\alpha} \sum_{\beta=1}^{d+d'} x_{j\beta}'\theta_{\beta\alpha} \right)$$

$$= \sum_{\alpha=1}^{d+d'} \left( \left( \sum_{\beta=1}^{d} x_{i\beta}'\omega_{\beta\alpha} + \sum_{\beta=d+1}^{d+d'} x_{i\beta}'\omega_{\beta\alpha} \right) \left( \sum_{\beta=1}^{d} x_{j\beta}'\theta_{\beta\alpha} + \sum_{\beta=d+1}^{d+d'} x_{j\beta}'\theta_{\beta\alpha} \right) \right)$$

$$= \sum_{\alpha=1}^{d+d'} \left( \sum_{\beta=1}^{d} x_{i\beta}'\omega_{\beta\alpha} \sum_{\beta=1}^{d} x_{j\beta}'\theta_{\beta\alpha} + \sum_{\beta=d+1}^{d+d'} x_{i\beta}'\omega_{\beta\alpha} \sum_{\beta=1}^{d} x_{j\beta}'\theta_{\beta\alpha} \right.$$

$$\left. + \sum_{\beta=1}^{d} x_{i\beta}'\omega_{\beta\alpha} \sum_{\beta=d+1}^{d+d'} x_{j\beta}'\theta_{\beta\alpha} + \sum_{\beta=d+1}^{d+d'} x_{i\beta}'\omega_{\beta\alpha} \sum_{\beta=d+1}^{d+d'} x_{j\beta}'\theta_{\beta\alpha} \right)$$

$$= \sum_{\alpha=1}^{d+d'} \left( \sum_{\beta=1}^{d} x_{i\beta}'\omega_{\beta\alpha} \sum_{\beta=1}^{d} x_{j\beta}'\theta_{\beta\alpha} \right) + \sum_{\alpha=1}^{d+d'} \left( \sum_{\beta=d+1}^{d+d'} x_{i\beta}'\omega_{\beta\alpha} \sum_{\beta=1}^{d} x_{j\beta}'\theta_{\beta\alpha} \right)$$

$$+ \sum_{\alpha=1}^{d+d'} \left( \sum_{\beta=1}^{d} x_{i\beta}'\omega_{\beta\alpha} \sum_{\beta=d+1}^{d+d'} x_{j\beta}'\theta_{\beta\alpha} \right) + \sum_{\alpha=1}^{d+d'} \left( \sum_{\beta=d+1}^{d+d'} x_{i\beta}'\omega_{\beta\alpha} \sum_{\beta=d+1}^{d+d'} x_{j\beta}'\theta_{\beta\alpha} \right)$$

Recall that $X_i'$ is the concatenation of $X_i$ and $T^l$. That means, for $1 \le \beta \le d$, $x_{i\beta}' \in X_i = \{x_{i\gamma}\}_{\gamma=1}^{d}$ and for $d+1 \le \beta \le d+d'$, $x_{i\beta}' \in T^l = \{\tau_\gamma(l)\}_{\gamma=1}^{d'}$. Furthermore, we decompose $W_q'$ as concatenation of two matrices $W_q, \tilde{W}_q$ such that $W_q = [\omega_{ij}]_{i,j=1,1}^{d,d+d'}$ and $\tilde{W}_q = [\omega_{ij}]_{i,j=d+1,1}^{d+d,d+d'}$. Similarly, we decompose $W_k'$ into $W_k$ and $\tilde{W}_k$. Then the previous expression for $a_{ij}'$ can be re-written as:

$$a_{ij}' = \sum_{\alpha=1}^{d+d'} \left( \sum_{\gamma=1}^{d} x_{i\gamma}\omega_{\gamma\alpha} \sum_{\gamma=1}^{d} x_{j\gamma}\theta_{\gamma\alpha} \right) + \sum_{\alpha=1}^{d+d'} \left( \sum_{\gamma=1}^{d'} \tau_\gamma(l)\omega_{\gamma+d,\alpha} \sum_{\gamma=1}^{d} x_{j\gamma}\theta_{\gamma\alpha} \right)$$

$$+ \sum_{\alpha=1}^{d+d'} \left( \sum_{\gamma=1}^{d} x_{i\gamma}\omega_{\gamma\alpha} \sum_{\gamma=1}^{d'} \tau_\gamma(l)\theta_{\gamma+d,\alpha} \right) + \sum_{\alpha=1}^{d+d'} \left( \sum_{\gamma=d+1}^{d'} \tau_\gamma(l)\omega_{\gamma+d,\alpha} \sum_{\gamma=d+1}^{d'} \tau_\gamma(l)\theta_{\gamma+d,\alpha} \right)$$

$$= (X_iW_q)(X_jW_k)^\top + (X_iW_q)(T^l\tilde{W}_k)^\top + (T^l\tilde{W}_q)(X_jW_k)^\top + (\tilde{W}_q\tilde{W}_k)(T^l \odot T^l)$$

$$= a_{ij} + A_{1i}T^{l\top} + T^lA_{2j} + A_3(T^l \odot T^l)$$

where $A_{i1}$, $A_{2j}$, and $A_3$ are $d'$ dimensional vectors corresponding the given input vector $X_i$. For input vector sequence $\mathbf{X}_i$, these form the time-evolution operators of attention, $\mathbf{A}_1, \mathbf{A}_2, A_3$.

# B  Properties of random sine-cosine matrices

In Section 5, we redesigned a single feed-forward operation at depth $l$ on a given input $X_i \in \mathbb{R}^d$ to produce output $X_{i+1} \in \mathbb{R}^{d'}$ as $X_{i+1} = \sigma(U^l \Sigma V^l X_i + B)$ where $U^l \in \mathbb{R}^{d \times d}$, $V^l \in \mathbb{R}^{d' \times d'}$ are random sine-cosine matrices to approximate rotation, $\Sigma \in \mathbb{R}^{d \times d'}$ is a rectangular diagonal matrix with learnable entries $\{\lambda_j\}_{j=1}^{min(d,d')}$, $B \in \mathbb{R}^{d'}$ is a learnable bias, and $\sigma(\cdot)$ is a non-linearity (ReLU in our case). $U^l$ ($V^l$) is defined as

$$U^l = \frac{1}{\sqrt{d}} \begin{bmatrix} \sin(w_{11}^l \frac{l}{P}) & \dots & \sin(w_{1\frac{d}{2}}^l \frac{dl}{2P}) & \cos(w_{11}^l \frac{l}{P}) & \dots & \cos(w_{1\frac{d}{2}}^l \frac{dl}{2P}) \\ \vdots & & & & & \vdots \\ \sin(w_{d1}^l \frac{l}{P}) & \dots & \sin(w_{d\frac{d}{2}}^l \frac{dl}{2P}) & \cos(w_{d1}^l \frac{l}{P}) & \dots & \cos(w_{d\frac{d}{2}}^l \frac{dl}{2P}) \end{bmatrix}$$

where $w_{ij}^l \in \mathcal{N}(0, \sigma^2)$ and $P = \frac{dL}{2\pi}$.

Let $A = U^l (U^l)^\top = [\alpha_{ij}]_{i,j=1,1}^{d,d}$. Then for all $1 \le i \le d$,

$$\alpha_{ii} = \sum_{j=1}^{\frac{d}{2}} \frac{1}{d} \left( \sin^2(w_{ij} \frac{jl}{P}) + \cos^2(w_{ij} \frac{jl}{P}) \right) = \frac{1}{2}$$

For all $i \ne j$,

$$\alpha_{ij} = \frac{1}{d} \sum_{k=1}^{\frac{d}{2}} \left( \sin(w_{ik} \frac{kl}{P}) \sin(w_{jk} \frac{kl}{P}) + \cos(w_{ik} \frac{kl}{P}) \cos(w_{jk} \frac{kl}{P}) \right)$$

$$= \frac{1}{d} \sum_{k=1}^{\frac{d}{2}} (A_k + B_k)$$

where $A_k = \sin(w_{ik} \frac{kl}{P}) \sin(w_{jk} \frac{kl}{P})$ and $B_k = \cos(w_{ik} \frac{kl}{P}) \cos(w_{jk} \frac{kl}{P})$. Let $\frac{kl}{P} = \kappa$; then we can rewrite $A_k$ and $B_k$ as:

$$A_k = \left( \frac{\exp(\mathbf{i} w_{ik} \kappa) - \exp(-\mathbf{i} w_{ik} \kappa)}{2\mathbf{i}} \right) \left( \frac{\exp(\mathbf{i} w_{jk} \kappa) - \exp(-\mathbf{i} w_{jk} \kappa)}{2\mathbf{i}} \right)$$

$$= \frac{-1}{4} \left( \exp(\mathbf{i} w_{ik} \kappa + \mathbf{i} w_{jk} \kappa) + \exp(-\mathbf{i} w_{ik} \kappa - \mathbf{i} w_{jk} \kappa) \right.$$
$$\left. - \exp(\mathbf{i} w_{ik} \kappa - \mathbf{i} w_{jk} \kappa) - \exp(-\mathbf{i} w_{ik} \kappa + \mathbf{i} w_{jk} \kappa) \right)$$

$$B_k = \left( \frac{\exp(\mathbf{i} w_{ik} \kappa) + \exp(-\mathbf{i} w_{ik} \kappa)}{2} \right) \left( \frac{\exp(\mathbf{i} w_{jk} \kappa) + \exp(-\mathbf{i} w_{jk} \kappa)}{2} \right)$$

$$= \frac{1}{4} \left( \exp(\mathbf{i} w_{ik} \kappa + \mathbf{i} w_{jk} \kappa) + \exp(-\mathbf{i} w_{ik} \kappa - \mathbf{i} w_{jk} \kappa) \right.$$
$$\left. + \exp(\mathbf{i} w_{ik} \kappa - \mathbf{i} w_{jk} \kappa) + \exp(-\mathbf{i} w_{ik} \kappa + \mathbf{i} w_{jk} \kappa) \right)$$

Assuming $w_{ik} \in X$ and $w_{jk} \in Y$ where $X$ and $Y$ are two independent random variables with pdf defined as $f(X) = \frac{1}{\sigma \sqrt{2\pi}} \exp(-\frac{X^2}{2\sigma^2})$ and $f(Y) = \frac{1}{\sigma \sqrt{2\pi}} \exp(-\frac{Y^2}{2\sigma^2})$,

$$\mathbb{E}[\exp(\mathbf{i} w_{ik} \kappa + \mathbf{i} w_{jk} \kappa)] = \frac{1}{2\pi\sigma^2} \int_{-\infty}^{\infty} \int_{-\infty}^{\infty} \exp(\mathbf{i} X\kappa + \mathbf{i} Y\kappa) \exp(-\frac{X^2}{2\sigma^2}) exp(-\frac{Y^2}{2\sigma^2}) dX dY$$

$$= \exp(-\frac{\sigma^2}{2} \kappa)$$

$$= \mathbb{E}[\exp(\mathbf{i} w_{ik} \kappa - \mathbf{i} w_{jk} \kappa)] = \mathbb{E}[\exp(-\mathbf{i} w_{ik} \kappa - \mathbf{i} w_{jk} \kappa)]$$

Then

$$\mathbb{E}[A_k] = \frac{-1}{4} \left( 2 \exp(-\frac{\sigma^2}{2} \kappa) - 2 \exp(-\frac{\sigma^2}{2} \kappa) \right) = 0$$

and similarly,

$$\mathbb{E}[B_k] = \frac{1}{4} \left( 2 \exp(-\frac{\sigma^2}{2} \kappa) + 2 \exp(-\frac{\sigma^2}{2} \kappa) \right) = \exp(-\frac{\sigma^2}{2} \kappa)$$

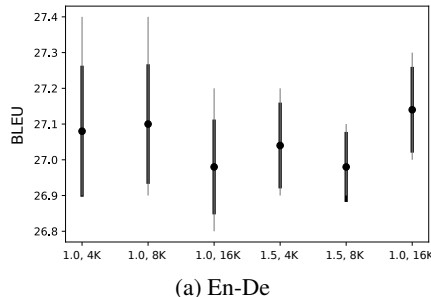

(a) En-De

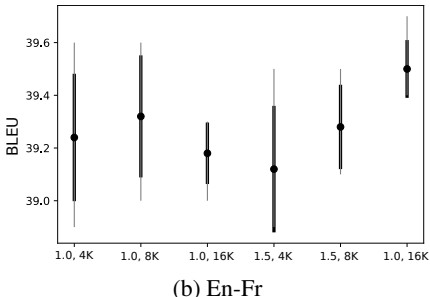

(b) En-Fr

Figure 2: Variation of BLEU score for En-De (WMT 2014) and En-Fr (WMT 2014) translation with different learning rates and warmup steps. x-axis in both plots show the $(lr_{max}, warmup\_step)$ pairs. The model variation used here in `TransEvolve`-**fullFF**.

Therefore, $\mathbb{E}[\alpha_{ij}] = \frac{1}{d} \sum_{k=1}^{\frac{d}{2}} \exp(-\frac{\sigma^2}{2} \frac{kl}{P})$ which approaches 0 as $\sigma$ gets larger. Thus, on the limiting case, we get $\mathbb{E}[U^l (U^l)^\top] = \frac{1}{2} \mathbf{I}_d$ where $\mathbf{I}_d$ is the $d$-dimensional identity matrix. This way, $U^l$ approximates a rotation matrix as we choose $\sigma = \mathcal{O}(d)$.

## C  Task related details

Here we describe the experimental details for encoder-decoder and encoder-only tasks. `TransEvolve` is implemented using Tensorflow version 2.4.1.

**Machine translation.**  For both En-De and En-Fr tasks, we use a batch size of $512$ with maximum allowed input sentence length of $256$ while training and train for a total of $300,000$ steps. Time needed for training varies with model configurations: `TransEvolve`-**randomFF**-1 takes 18 hours to finish while `TransEvolve`-**fullFF**-2 takes around 32 hourrs. All of these training and testings are done with 32-bit floating point precision. To find the optimal learning rate, we used the following pairs of $(lr_{max}, warmup\_step)$ values (see Section 7.3): $(1.0, 4000)$, $(1.0, 8000)$, $(1.0, 16000)$, $(1.5, 4000)$, $(1.5, 8000)$, and, $(1.5, 16000)$. For all the experiments, the optimizer we use is Adam with $\beta_1 = 0.9$, $\beta_2 = 0.98$, and $\epsilon = 10^{-9}$. We used beam search with beam size 4 and length penalty 0.6. For En-De task, we used an extra decode length of 50; for En-Fr, this value is set to 35. Figure 2 summarizes the variation in performance with different $(lr_{max}, warmup\_step)$ values; we run 5 independent training and testing with different random seeds, and choose the maximum BLEU score from each runs to plot this variation.

**Encoder-only tasks.**  As mentioned in Section 7.1, we experiment with the small version of `TransEvolve` variants ($d = 256$) for all the encoder-only tasks. We set the values of $(lr_{max}, warmup\_step)$ to $(0.5, 8000)$ and use the default parameters of Adam to optimize. All encoder-only experiments are done using a maximum input length of $512$.

In the text classification regime, we use the BERT (base uncased) tokenizer from Huggingface[1]. The batch size is set to $80$. We train each model for $15$ epochs. However, the best models emerge by 7-8 epochs of training with a $\pm 0.2\%$ error range in test accuracy over 5 randomly initialized runs.

In the long range sequence classification regime, the tokenization (character-level in IMDB and operation symbols in ListOps) and maximum input lengths are predefined . We use a batch size of $48$ for the IMDB dataset, and $64$ for the ListOps dataset. Again, we train all the models for $15$ epochs, with best performances emerging after 9-10 epochs of training with error margins $\pm 0.8\%$ in ListOps and $\pm 0.3$ in IMDB datasets.

---

[1]`https://huggingface.co/transformers/model_doc/bert.html#berttokenizer`