# OpenReview forum: "Redesigning the Transformer Architecture with Insights from Multi-particle Dynamical Systems"
_NeurIPS.cc/2021/Conference — NeurIPS 2021 Spotlight_

### Official Review · Reviewer_2RYY · 2021-07-14

**Rating:** 6
**Confidence:** 3

**Summary:**

This paper avoid computing pairwise dot-product between n input vectors at each layer by computing a functional form at the  beginning and evolving it in a temporal (depth-wise) manner.

**Limitations And Societal Impact:**

See main review

**Main Review:**

Major questions
1.Why this point of view relates to the ODE viewpoint, can you formally written in an ODE setting to show the two models have the similar

2. I believe [1] is highly related to your paper, I think it would be essential to discuss and compare with it to show the dynamically improving the attention weights is ensential.

Minor points
The notation in the paper is somehow confusing, I would suggest the authors to organize the presentation in next version, there are too many notations doesn't defined in the section 4.

I'm not a NLP guy, thus I would like to let other reviewers to evaluate the experiments, in my limited experience, the experiments looks good

From my perspective, this is a very good direction to reduce the quadratic computation power for transformer , the drawback is the unclear motivation from the dynamic system viewpoint to the proposed structure.

[1] Ying C, Ke G, He D, et al. LazyFormer: Self Attention with Lazy Update[J]. arXiv preprint arXiv:2102.12702, 2021.

**Time Spent Reviewing:**

0.75

---

> ### Author Response · Authors · 2021-08-10
> **Response to the initial reviews from Reviewer 2RYY**
>
> “Why this point of view relates to the ODE viewpoint, can you formally written in an ODE setting to show the two models have the similar”
>
> Response: The underlying ODE in TransEvolve is the same as Transformer, as shown in Equation 4, and the numerical approximation in Equation 5  in the main text. We explained in lines 113-119 that while Transformer seeks to learn the functions F and G in Equation 5 at each step (depth) independently, TransEvolve defines temporal (depthwise) evolution functionals to realize them that take the initial conditions and time (depth) as inputs. The underlying ODE in TransEvolve is the same as Transformer, as shown in Equation 4. Transformer seeks to design the numerical  approximation as:
> $$\tilde{x}_i(t) = x_i(t) + \delta tF(x_i(t), \mathbf{x}(t), t) \approx x_i(t) + \mathcal{F}(x_i(t), \mathbf{x}(t))$$
>
> $$x_i(t+\delta t) = \tilde{x}_i(t) + \delta tG(\tilde{x}_i(t), t) \approx \tilde{x}_i(t) + \mathcal{G}(\tilde{x}_i(t))$$
> Here $t$ is removed from Equation 5. That is, at any $(t, t+\delta t]$ interval, there is no information about the functions $ \mathcal{F}$  and $\mathcal{G}$ modeled at previous intervals. Subsequently, the inter-particle interactions (i.e., attention) and independent particle evolution (i.e., pointwise feed-forward) need to be recomputed at each layer without the knowledge of their past counterparts. TransEvolve redefines Equation 5 to incorporate the initial state of the ODE $ \mathbf{s} = [s_i] =\mathbf{x}(t_0)$ and time $t$ (depth) into account:
> 	$$\tilde{x}_i(t) = x_i(t) + \delta tF(x_i(t), \mathbf{x}(t), t) \approx x_i(t) + \tilde{\mathcal{F}}(x_i(t), f(\mathbf{s}, t))$$
> $$x_i(t+\delta t) = \tilde{x}_i(t) + \delta tG(\tilde{x}_i(t), t) \approx \tilde{x}_i(t) + \mathcal{G}(\tilde{x}_i(t), g(t))$$
> In the case of randomFF versions, the $g(t)$ is modeled using non-parametric functions of t, while fullFF versions keep them in their original parametric versions. Both of these versions, however, use the same strategy to model $f(\mathbf{s}, t)$ as described in Section 4 and Appendix A.
>
> “I believe [1] is highly related to your paper, I think it would be essential to discuss and compare with it to show the dynamically improving the attention weights is ensential.”
>
> Response: Thanks for bringing this to notice. We will certainly discuss LazyFormer in relation to TransEvolve. LazyFormer computes the self-attention weights from embedding layer outputs and reuses them in subsequent layers. That said, the attention values do not change over different depths, i.e., the attention between the i-th query and the j-th key remains in any subsequent layer. As several earlier, probing experiments on deep transformer models [2, 3] suggest, they do learn to put different attention weights among the same pairs of the input tokens. Moreover, these weights can often be compared to several linguistic structures, i.e., syntactic dependency, next tokens, periods, etc. [3]. TransEvolve, while reusing initial dot-product attention in subsequent layers, also defines a depth-wise evolution scheme to achieve this variety. Given the short amount of time for author response, we ran encoder-only experiments with LazyFormer 2x3 (i.e., 3 blocks of size 2) on the AGNews and ListOps datasets. We got 89.46% and 36.39% accuracy on AGnews and ListOps, respectively. Both these performances are lower than TransEvolve (91.1% and 43.2%, respectively) at a higher parameter cost.
>
> “The notation in the paper is somehow confusing”
>
> Response: We will look into this and modify the final version accordingly. We will add a separate notation table.
>
> [1] Ying C, Ke G, He D, et al. LazyFormer: Self Attention with Lazy Update[J]. arXiv preprint arXiv:2102.12702, 2021.
>
> [2] Paul Michel, Omer Levy, and Graham Neubig. Are sixteen heads really better than one? In Advances in Neural  Information  Processing  Systems 32:   Annual  Conference on  Neural  Information  Processing Systems  2019.
>
> [3] Kevin Clark, Urvashi Khandelwal, Omer Levy, and Christopher D. Manning. What does BERT look at? an analysis of BERT’s attention. In BlackboxNLP@ACL, 2019.

---

> ### Author Response · Authors · 2021-08-24
> **Following up rebuttal response**
>
> Dear reviewer,
> We have tried to address your concerns in our rebuttal. We have explained the ODE point-of-view of the attention evolution. Also, we have added experiments related to LazyFormer as you suggested. Please let us know your opinion regarding this.

---

> > ### Comment · Reviewer_2RYY · 2021-08-27
> > **Sorry for the reply**
> >
> > I'm still not convinced by the neccesity of the dynamic system viewpoint, although I'm ok with a story with the ode viewpoint. The approximation have nothing to do with the ode.
> >
> > The experiment relates to the Lazyformer is good, it demonstrated the benefit of the paper.
> >
> > At this point due to the hardness to read this version of paper and the connection and not so cleared intuition,  I'll keep my score at 6.
> >
> > I looked at other review, i think this paper is worth to be accepted.

---

### Official Review · Reviewer_8Tqc · 2021-07-16

**Rating:** 7
**Confidence:** 3

**Summary:**

The authors introduce the TransEvolve approach based on the reformulation of transformer architectures in terms of the temporal evolution of many particle systems. Doing so, they can significantly reduce model parameters and computational cost, while achieving competitive or better performance than existing methods on a series of benchmark tasks.

**Limitations And Societal Impact:**

Potential limitations and societal impact have been sufficiently addressed.


**Main Review:**

The use of ODE formalism to cast the transformer into a dynamic framework is an interesting concept, at it allows to formulate multi-head attention and feed forward operations as the evolution of time series, eliminating the need for several costly computations and reducing the total number of parameters.

The experiments are performed on a series of benchmarks covering a range of potential applications. In almost all tasks, at least one TransEvolve variant improves upon the performance of other models demonstrating the utility of the approach. However, here the text could benefit for a more in-depth discussion on the reasons for the performance difference observed between the random matrix and full feed forward variants.

Overall the paper is written in a clear manner, the architecture and development of the model is easy to follow and results are presented in a concise fashion.

TransEvolve offers a different view on transformer architectures. The theoretical connections to dynamical systems could be exploited to better analyze and potentially improve these kinds of models.

**Time Spent Reviewing:**

3

---

> ### Author Response · Authors · 2021-08-10
> **Response to the initial reviews from Reviewer 8Tqc**
>
> “However, here the text could benefit for a more in-depth discussion on the reasons for the performance difference observed between the random matrix and full feed forward variants.”
>
> Response: Thank you for the valuable suggestion. We will add more intuitive explanations for this observation. In line 342-344, we have already mentioned that in Machine Translation (MT), the total number of parameters does play a role in performance (hence, random versions with fewer parameters are outperformed by the original full versions). The way we designed the random matrix feedforward transformations incorporates the depth information via evolution while reducing the number of parameters. In encoder-only tasks, this evolution scheme looks comparable to depth-independent, parameter-dense full versions in expressive power. Moreover, each of the tasks presents different learning requirements. Intuitively, ListOps or character-level sentiment classification tasks have a smaller input vocabulary (hence, fewer amounts of information to encode in each embedding), but longer intra-token dependency (resulting in a need for a powerful attention mechanism) compared to sentiment or topic classification at the word level. Random matrix versions provide depth-wise information sharing that may facilitate the model to better encode complex long-range dependencies; but they might remain under-parameterized to transform information-rich hidden representations. This complexity trade-off can be the possible reason behind randomFF versions outperforming fullFF in both the long-range tasks while vice versa in text classification tasks. We will add this additional discussion in the final version.

---

> > ### Comment · Reviewer_8Tqc · 2021-08-20
> > **Response to rebuttal**
> >
> > Thank you for the additional explanations and proposed revisions to the text. My rating remains at 7.

---

### Official Review · Reviewer_YDLp · 2021-07-19

**Rating:** 7
**Confidence:** 3

**Summary:**

This paper introduces TransEvolve, a new efficient transformer architecture that approximates the multi-headed self-attention and point-wise feed-forward transformations in the standard transformers.  Drawing from recent work by Lu et al., 2019 which established the connection between transformers and modeling the dynamics of a multi-particle dynamic systems, TransEvolve approximate multiple layers of the transformer with a temporal evolution scheme that bypasses costly dot product operations.  Experiments on encoder-decoder tasks show TransEvolve to be competitive with vanilla transformers while being slightly more efficient.  On encoder-only tasks, TransEvolve yields better predictive performance with half as many parameters.

**Limitations And Societal Impact:**

The authors sufficiently discuss the computational limitations of TransEvolve compare to existing approaches.

**Main Review:**

## Originality
The connection between transformers and multi-particle dynamical systems was established in Lu et al., 2019.  This paper leverages this connection to inform the design of more efficient transformer architectures.  The main points of novelty in this work is the design temporal approximation schemes for the multi-headed attention and point-wise feedforward components of transformers architectures.

## Quality
The overall quality of the paper is sufficient.  I do think the experimental results can benefit from additional discussion to provide insights into why TransEvolve performs much better than existing methods for encoder-only tasks. I found L341-354 interesting and believe the paper would benefit greatly from additional discussion to help readers gain more intuition for how TransEvolve behaves.

Questions:
- Did you try other values of L besides the two settings?  How does TransEvolve perform with even higher L > 6 or lower cumulative L < 6?
- Lu et al., 2019 introduced an architecture called Macaron Net motivated by a different splitting scheme.  Did you look into approximations for that architecture? It seems like  Macaron Net does outperform vanilla transformers on WMT14 by a healthy margin.


## Clarity
The paper is well written overall but I would have benefited from more intuitive explanation of the approximations presented in Sections 4 and 5.  I did appreciate the complexity summaries provided for the two approaches in the end of each section.

_Clarification needed:_
- L343-346: "In the case of machine translation though, it is straightforward that having a lesser number of feed-forward parameters and/or fewer encoder blocks with deeper evolution deteriorate the performance" this seems to indicate that performance increases with parameter size.  But the next sentence states "In general, the performance of TransEvolve variations changes inversely with the total parameter size on translation tasks." Am I misinterpreting something here?

_Typos:_
- L349-350: "learn more useful representations of the input with a much fewer number of trainable parameters" -> learn more useful representations of the input with much fewer trainable parameters


## Significance
The contributions of the paper are important owing to the new perspective provided through the temporal approximations derived from taking a multi-partical dynamical systems view of multi-layer attention.  That said, I am unsure how useful TransEvolve will be in practice given the apparent need for careful tuning (i.e. there isn't a setting from randomFF-1, randomFF-2, fullFF-1, and fullFF-2 that worked well across all experiments) and the mixed experimental results (i.e. TransEvolve seems to only improve upon existing approaches in the encoder-only setting).  Additionally, as the authors noted, TransEvolve is still less efficient than linear attention models since it incurs quadratic cost to compute the initial states of the system.


===========Post Author Response===========
After reading the author's response, I have decided to change my score to 7.  I strongly encourage the authors to include the intuitive explanation of why TransEvolve performs better on encoder only tasks and discussion of potential directions for future work in the paper.

**Time Spent Reviewing:**

4.5

---

> ### Author Response · Authors · 2021-08-10
> **Response to the initial reviews from Reviewer YDLp**
>
> “I do think the experimental results can benefit from additional discussion to provide insights into why TransEvolve performs much better than existing methods for encoder-only tasks.”
>
> Response: Thank you very much for the valued suggestion. We will add some intuitive explanations. Encoder-only models use only self-attention, while the encoder-decoder models use self, cross, and causal attentions. From the viewpoint of multi-particle dynamical systems, the latter two are somewhat different from the former one. In self-attention, each of the ‘particles’ is interacting with each other simultaneously at a specific time-step. However, in causal attention, the interactions are ordered based on the token positions even within a specific timestep. So while decoding, there is an additional evolution of token representations at each step. Moreover, in encoder-decoder tasks as well, TransEvolve outperforms the original Transformer in two datasets (En-De 2013 and En-Fr 2014) clearly, while providing comparable performance in another (En-De 2014). It is the random matrix versions that suffered the most in these versions. The way we designed the random matrix feedforward transformations incorporates the depth information via evolution while reducing the parameter size. In encoder-only tasks, this evolution scheme looks comparable to depth-independent, parameter-dense full versions in expressive power. Moreover, each of the tasks presents different learning requirements. Intuitively, ListOps or character-level sentiment classification tasks have a smaller input vocabulary (hence, fewer amounts of information to encode in each embedding), but longer intra-token dependency (resulting in a need for a powerful attention mechanism) compared to sentiment or topic classification at the word level. Random matrix versions provide depth-wise information sharing that may facilitate the model to better encode complex long-range dependencies, but they might remain under-parameterized to transform information-rich hidden representations. This complexity trade-off can be the possible reason behind randomFF versions, outperforming fullFF in both the long-range tasks while vice versa in text classification tasks. We will add this additional discussion in the final version.
>
> “Did you try other values of L besides the two settings? How does TransEvolve perform with even higher L > 6 or lower cumulative L < 6?”
>
> Response: Yes, we experimented with varying cumulative values of L for the WMT tasks. We observed that the performance degrades for all the settings with L<6. With L>6, random feedforward versions show no significant improvements while full models either remain the same or degrade. This is similar to the findings of Levine et al. [1].
>
> “Lu et al., 2019 introduced an architecture called Macaron Net motivated by a different splitting scheme. Did you look into approximations for that architecture?”
>
> Response: Yes, we experimented with the MacaronNet version (sandwiching the attention layers with feed-forward blocks instead of a single feedforward block following the attention) in TransEvolve. The performance degraded in these experiments (1.2 and 1.5 points drop on En-De 2014 and En-Fr 2014 development data, respectively, compared to TransEvolve, without beam search). MacaronNet claims to achieve a second-order approximation of the underlying ODE by using the Strang-Marchuk splitting scheme instead of Lie-Trotter. However, the complete numerical approximation is still being done using Euler’s method which is a first-order approximation. This looked theoretically confusing to us.
>
> “I would have benefited from more intuitive explanation of the approximations presented in Sections 4 and 5.”
>
> Response: Thank you for your kind suggestion. Sure, we will add more intuitive explanations in the modified version as explained in Response #1.
>
> “Clarification needed: L343-346”
>
> Response: Thanks for pointing it out. The line should read “In general, the performance of TransEvolve variations changes proportionally with the total number of parameters on translation tasks.” We will incorporate this in our final version.
>
> “I am unsure how useful TransEvolve will be in practice given the apparent need for careful tuning”
>
> Response: At a high level, TransEvolve works like many recent techniques to mitigate Transformer’s high computational complexity by avoiding full self-attention. All such methods need some amount of per-application tuning to maximize benefits, but this is amortized by many runs of much faster training. Our experiments suggest that in Machine Translation, the full versions tend to outperform the randomized ones; while in encoder-only tasks, they provide comparable results. We believe the general family of approaches is well motivated from the time-accuracy tradeoff perspective, and we can therefore focus on comparisons within the family. RFA and Performer use almost similar methods of kernel approximation to linearize attention, the only difference being trigonometric kernels in RFA as opposed to positive random features in Performer. There is no clear winner among these two similar methods over different tasks.
>
> [1] Yoav  Levine,  Noam  Wies,  Or  Sharir,  Hofit  Bata,  and  Amnon  Shashua.   Limits to depth efficiencies of self-attention. In Advances in Neural Information Processing Systems 33: Annual Conference on Neural Information Processing Systems 2020

---

> ### Author Response · Authors · 2021-08-24
> **Following up rebuttal response**
>
> Dear reviewer,
> We have tried to address your concerns in our rebuttal. We have added additional experimental observations as you suggested. Additional intuitive explanations are proposed. Also, we have given the clarifications you asked for. Please let us know your comments.

---

> > ### Comment · Reviewer_YDLp · 2021-08-26
> > **Post author response**
> >
> > Thank you for responding to the points in my review.  Could you comment on the potential directions for future work that build on top of TransEvolve?  Also, given the still quadratic dependency of TransEvolve, do you think it's fruitful to explore ways of removing the quadratic cost with some sort of linear approximation?

---

> > > ### Author Response · Authors · 2021-08-26
> > > **Comments on potential future work**
> > >
> > > Linear approximation: Among the recent endeavors in linearizing Transformer attention, random feature methods have gained substantial progress. Typically, these methods seek to approximate $\operatorname{Softmax}(\mathbf{q}^\top\mathbf{K})$ as $\phi(\mathbf{q})^\top\phi(\mathbf{K})$ where $\phi()$ is the random feature map. Following the time-evolution property introduced in TransEvolve, a natural choice of design would then be a time-evolution of $\phi$ as $\tilde{\phi}(\cdot, t)$.
> > >
> > > Possible future work: The ODE point-of-view of sequence learning, as introduced in TransEvolve, opens some interesting areas of possible future work. The possible linearization with time-evolution has already been pointed out. Another possible line of investigation can be to incorporate higher-order numerical approximation of ODEs into play (for example, Adam-Bashforth method instead of Euler). Moreover, as we pointed out in lines 350-356, an in-depth theoretical analysis is a possible future work.

---

### Official Review · Reviewer_N59G · 2021-08-02

**Rating:** 8
**Confidence:** 4

**Summary:**

This paper develops an analogy between Transformer networks and dynamical systems of multiple interacting particles. They follow the work of Lu et al. 2019 using Lie-Trotter splitting scheme to derive differential equations approximating terms corresponding to operations that compose the Transformer network internals, namely multi-head self-attention and feed-forward transformation.

This paper develops the described technique beyond previous work by noting their analogous dynamical system motivates architecture choices that would require many fewer parameters than the associated Transformer. The authors investigate performance and memory gains by exploiting time-dependence in the dynamical transformation, achieving similar results to analogous weight-sharing between layers.

**Ethical Concerns:**

No ethical concerns

**Limitations And Societal Impact:**

The authors clearly reiterate their technical contributions relative to existing Transformer networks. Primarily, they address the limitations of their complexity reductions, e.g. not affecting the quadratic cost in initial conditions. The authors do not discuss potential negative societal impact. As basic research it is difficult to suggest improvement here.

**Main Review:**

This work builds on existing literature that relates familiar deep-learning structures to solutions of corresponding dynamical systems. The Lie-Trotter splitting scheme in particular establishes the connection to Transformer networks in literature. These previously published results are well-cited and presented in this paper. As well, this work develops these techniques beyond previously published methods by investigating various parameter sharing formulations that are naturally motivated by their dynamical system. They show improvements in complexity and demonstrate practically through experimental results.

**Time Spent Reviewing:**

5

---

> ### Author Response · Authors · 2021-08-10
> **Response to the initial reviews from Reviewer N59G**
>
> Thank you very much for your valuable review. We are grateful that you liked our paper.

---

### Decision · Program_Chairs · 2021-09-27

**Decision:**

Accept (Spotlight)

**Comment:**

This paper was universally well-liked by the reviewers. The reviewers found the theoretical developments to be elegant, with significant practical applications yielding a convincing new model in the transformer family. Several reviewers commented that the exposition might be improved by including more intuitive explanations.